# Association of serum 25-hydroxyvitamin D levels with severe necroinflammatory activity and inflammatory cytokine production in type I autoimmune hepatitis

**Kazumichi Abe** [ID]*, **Masashi Fujita, Manabu Hayashi, Atsushi Takahashi, Hiromasa Ohira**

Department of Gastroenterology, Fukushima Medical University, Fukushima, Japan

* k-abe@fmu.ac.jp

## Abstract

25-Hydroxyvitamin D [25(OH)D] has been reported to be associated with several chronic liver diseases. The relationship between 25(OH)D and autoimmune hepatitis (AIH) pathogenesis is incompletely understood. We investigated the association of serum total and free 25(OH)D levels with necroinflammatory activity and cytokine levels in 66 patients with AIH diagnosed in our hospital. The median age at AIH diagnosis was 57 years, and the male: female ratio was 7:59. The median serum total 25(OH)D level in therapy-naïve patients with AIH was 14.2 ng/mL (interquartile range [IQR], 11.4–17.9 ng/mL). Of the 66 patients with AIH, 36 had serum total 25(OH)D levels of < 15 ng/mL and were considered to have vitamin D deficiency, and 30 had serum total 25(OH)D levels of $\geq$ 15 ng/mL. Patients with acute-onset AIH had significantly lower serum total 25(OH)D levels than those with chronic-onset AIH. In particular, serum total 25(OH)D levels were significantly lower in patients with severe forms of AIH. Furthermore, the serum total 25(OH)D level was positively correlated with the serum albumin level and prothrombin time and negatively correlated with the serum total bilirubin level and necroinflammatory activity in AIH. Multivariate logistic regression analysis showed that the serum total 25(OH)D level was an independent factor for severe necroinflammatory activity. Interestingly, AIH patients with serum total 25(OH)D levels of < 15 ng/mL had higher levels of inflammatory cytokines such as interferon-γ and interleukin-33. Free 25(OH)D levels were correlated with total 25(OH)D levels, and the percentage of free 25 (OH)D was significantly associated with necroinflammatory activity. In conclusion, 25(OH)D deficiency may play an important role in predicting AIH severity via inflammatory cytokine production.

## Introduction

Autoimmune hepatitis (AIH) presents as chronic hepatitis of unknown cause. AIH is commonly associated with the presence of autoantibodies and hypergammaglobulinemia. Histologically, interface hepatitis with marked lymphocyte and plasma cell infiltration is observed.

**Data Availability Statement:** All relevant data are within the manuscript and its Supporting information files.

**Funding:** This study was supported in part by AbbVie Inc. There was no additional external funding received for this study.

**Competing interests:** Kazumichi Abe and Hiromasa Ohira has received research funds from AbbVie Inc. All other authors declare no conflict of interest. This does not alter our adherence to PLOS ONE policies on sharing data and materials.

**Abbreviations:** 25(OH)D, 25-hydroxyvitamin D; AIH, autoimmune hepatitis; ALB, albumin; ALP, alkaline phosphatase; ALT, alanine aminotransferase; ANA, antinuclear antibody; AST, aspartate aminotransferase; CI, confidence interval; ELISA, enzyme-linked immunosorbent assay; H&E, hematoxylin and eosin; HBV, hepatitis B virus; HC, healthy control; IAIHG, International Autoimmune Hepatitis Group; Ig, immunoglobulin; IHC, immunohistochemical; IL, interleukin; LKM-1, liver/kidney microsomal 1; OR, odds ratio; PLT, platelet; PSL, prednisolone; PT, prothrombin time; TB, total bilirubin; Th, helper.

AIH is a disease that develops and progresses due to autoimmune reactions caused by a failure of immunological tolerance to hepatocytes, and it is speculated that various genetic and immunological abnormalities are involved. The incidence of AIH with acute presentation has recently increased, and this form has been established as type I or II AIH [1–6].

Vitamin D has received special attention because it plays an immunomodulatory role and blocks the inflammatory process by suppressing attacks by T and B cells. Reduced vitamin D levels are frequently found in patients with various autoimmune diseases [7]. The liver plays an important role in vitamin D synthesis. Vitamin D is hydroxylated by cytochrome P450 enzymes to produce 25-hydroxyvitamin D [25(OH)D]. 25(OH)D is considered the most reliable indicator of serum vitamin D levels [8]. Low serum vitamin D levels have been recognized in several chronic liver diseases [9,10].

Low 25(OH)D levels have been observed in 81% of patients with AIH [10]. Furthermore, vitamin D deficiency has been associated with increased mortality in patients with cirrhosis [11,12] and hepatocellular carcinoma [13] and with severe interface hepatitis, advanced fibrosis, and nonresponse to conventional immunosuppressive therapy in patients with AIH [10]. However, correlations of the vitamin D level with the inflammatory response and immune function of patients with AIH have rarely been studied.

Recently, the measurement of free 25(OH)D as a potentially more accurate marker of vitamin D status has attracted interest [14]. Total 25(OH)D is in equilibrium with the forms bound to vitamin D binding protein (VDBP) and to albumin (ALB) and is consequently dependent on the levels of both proteins; however, free 25(OH)D concentrations are independent of the level of either protein. The relationship between free 25(OH)D and AIH pathogenesis is not understood.

Cytokines are soluble factors that are released by immune cells and are subsequently involved in immune cell differentiation, maturation and activation [15]. AIH is a chronic hepatitis of unknown etiology, although several cytokines have been implicated in its pathogenesis and severity.

In this study, we investigated the association of serum total and free 25(OH)D levels with necroinflammatory activity and cytokine levels in patients with AIH.

## Methods

### Study population

The subject cohort comprised 66 patients with AIH diagnosed at Fukushima Medical University Hospital between 1986 and 2019. The diagnosis of AIH was based on the revised and simplified International Autoimmune Hepatitis Group (IAIHG) scoring system [16–18]. Autoantibodies (antinuclear antibodies [ANAs] and anti-smooth muscle antibodies [anti-SMAs]) were detected by indirect immunofluorescence in HEp-2 cells or frozen sections of rat kidney tissue. Serum anti-liver/kidney microsomal 1 (anti-LKM-1) antibody was detected by enzyme-linked immunosorbent assay (ELISA) using commercially available kits. Patients with chronic liver disease due to other causes, particularly alcohol abuse and chronic hepatitis B virus (HBV) or hepatitis C virus infection, were excluded from the AIH group. Patients with AIH were excluded if they tested positive for Epstein-Barr virus (EBV), cytomegalovirus (CMV), hepatitis A virus (HAV), or hepatitis E virus (HEV) as assessed by the presence of IgM-EBV antibody, IgM-CMV antibody, IgM-HAV antibody, or IgA-HEV antibody, respectively. In addition, patients with drug-induced liver injury (DILI) as assessed by the Digestive Disease Week Japan (DDW-J) 2004 scale, Japanese clinical criteria for DILI [19] and liver biopsy findings were excluded. Assessed data that were extracted retrospectively from the patients' medical charts included background characteristics (age, sex, onset type of disease,

and IAIHG score), clinical parameters at presentation (aspartate aminotransferase [AST], alanine aminotransferase [ALT], alkaline phosphatase [ALP], total bilirubin [TB], and ALB levels; platelet [PLT] count; prothrombin time [PT]; and IgG, ANA, anti-SMA and anti-LKM-1 titers) and at remission (ALT, ALP, TB, and IgG), presence of relapse, presence of cirrhosis, disease severity, and therapeutic methods. Acute AIH was defined by the presence of acute-onset symptoms (e.g., jaundice, fatigue and/or anorexia) in conjunction with a bilirubin level of >5 mg/dL and/or a serum ALT level of greater than 10-fold higher than the upper limit of normal. Serum 25(OH)D levels and laboratory values were analyzed in patients with chronic AIH and acute AIH at the time of first diagnosis. Relapse of AIH was defined as an increase in serum transaminases to greater than twice the upper limit of normal (ALT level >90 U/L). Biochemical remission was defined as the complete normalization of aminotransferases and IgG [20]. The median time to remission was 63 days (IQR, 43–108 days). Two patients did not achieve remission. Liver cirrhosis was diagnosed based on histological findings, radiological findings of cirrhosis by computed tomography or magnetic resonance imaging, and the presence of complications, such as portal hypertension. Demonstration of histological cirrhosis at subsequent liver tissue examination, radiological findings of cirrhosis, and development of portal hypertension or its complications in patients without these findings at presentation indicated progression to cirrhosis. Disease severity was assessed in accordance with the diagnosis and treatment guide for AIH in Japan [21]. Twenty patients had a severe form of AIH (TB >5.0 mg/dL and/or PT <40%). Serum samples were obtained from the 66 patients with AIH at the time of diagnosis and before immunosuppressive treatment. None of the AIH patients were taking vitamin D supplements when the serum 25(OH)D levels were measured. All serum samples were frozen and stored in multiple tubes at -20°C until analysis. The average time from collection of serum samples to liver biopsy was 10.3 ± 19.2 days. Although corticosteroids are used as the first-line therapy for AIH, approximately 10% of AIH patients in Japan are resistant to steroid therapy. Azathioprine, which was not covered by national health insurance in Japan until July 2018, is considered the second-line therapy for steroid-resistant patients.

## Ethics statement

All patients agreed to serum and histological testing and provided written informed consent. The study protocol conformed to the ethical guidelines of the Declaration of Helsinki and was approved for the use of opt-out consent by the ethics committee of Fukushima Medical University (no. 2427). This study was performed in accordance with relevant guidelines and regulations, and all patients and control subjects agreed to undergo serum and histological testing and have their blood stored for future research.

## Assessment of serum total and free 25(OH)D levels

Serum total 25(OH)D levels were analyzed with a Liaison chemiluminescent immunoassay (CLIA, Hitachi Chemical Diagnostics Systems Co., Ltd, Tokyo, Japan). Patients with a serum 25(OH)D level of <15 ng/mL were considered to be deficient in vitamin D [22,23]. Serum free 25(OH)D levels were analyzed with an ELISA kit according to the manufacturer's instructions (DIAsource, ImmunoAssays S.A., Belgium).

## Immunoassays

Serum concentrations of cytokines, such as interleukin (IL)-1β, IL-4, IL-6, IL-10, IL-17A, IL-17A/F, IL-17F, IL-21, IL-22, IL-23, IL-25, IL-31, IL-33, interferon (IFN)-γ, tumor necrosis

factor (TNF)-α and sCD40L, were measured using a Luminex Bio-Plex 200 system (Bio-Rad, Hercules, CA) according to the manufacturer's protocol.

## Histological evaluation

Sixty-one patients with AIH underwent ultrasound-guided liver biopsy. Five patients were excluded from the pathologic analysis; 2 were excluded because of cirrhosis, and the others because of poor health conditions, such as hepatic failure. Liver sections were stained with hematoxylin and eosin (H&E). Slides were coded and read by two pathologists blinded to the patient identity and history. Histological evaluation was performed according to the classification of Scheuer [24] and Desmet et al [25]. The grades for necroinflammatory activity (G) and stages of fibrosis (S) ranged from G0 to G4 and from S0 to S4, respectively.

## Statistical analysis

Continuous variables are presented as the medians (interquartile ranges [IQRs]). Differences were compared using the Mann-Whitney U test and Wilcoxon matched-pairs signed-rank test. Correlations between variables were assessed using Spearman's rank correlation coefficient. To determine the optimal 25(OH)D cutoff level that could distinguish between severe and nonsevere necroinflammatory activity, receiver operating characteristic curves were used. Cutoff levels for the parameters were set at the points closest to 100% sensitivity and specificity. Univariate and multivariate logistic regression analyses were performed to analyze the factors related to the necroinflammatory activity grade. All statistical analyses were performed using Prism 6.0 software (GraphPad Software, Inc.) and JMP Pro 13.1 (SAS Institute Inc., Cary, NC). $P < 0.05$ was considered to indicate a significant difference.

## Results

### Patient characteristics

The baseline characteristics of the patients with AIH are summarized in Table 1. Of the 66 patients with AIH, 61 tested positive for ANAs, and 61 tested negative for anti-LKM-1 antibodies. These patients were positive for ANAs and diagnosed with type I AIH. Acute-onset AIH accounted for 52% of the cases. Twenty patients (30%) had a severe form of AIH (TB >5.0 mg/dL and/or PT <40%) [26]; 10 patients (15%) had cirrhosis at diagnosis, and 3 patients (5%) developed cirrhosis. Four patients (6%) developed decompensation. Eleven patients (18%) experienced relapse during immunosuppressive treatment; of these 11 patients, 3 experienced relapse after immunosuppressive treatment withdrawal, and 8 experienced relapse during continuous immunosuppressive treatment. The necroinflammatory activity grades were as follows: G0 (0/61, 0%), G1 (5/61, 8.2%), G2 (28/61, 45.9%), G3 (25/61, 41.0%), and G4 (3/61, 4.9%). The stages of fibrosis were as follows: F0 (6/61, 9.8%), F1 (9/61, 14.8%), F2 (19/61, 31.1%), F3 (19/61, 31.1%), and F4 (8/61, 13.1%). Regarding therapy, most patients (89%) were treated with prednisolone (PSL), and 15 (23%) were treated with azathioprine in combination with PSL [26].

The clinical, laboratory and histological characteristics of the patients with AIH stratified by the serum total 25(OH)D level (< 15 vs. ≥15 ng/mL) are presented in Table 1. Serum ALB, TB, AST levels and PT were associated with total 25(OH)D levels of < 15 ng/mL, while no association was observed between serum 25(OH)D levels and age, sex, PLT count, ALT level, ALP level, IgG titer, ANA titer, AIH score, relapse frequency and cirrhosis. The number of patients treated with azathioprine in combination with PSL was higher among patients with

**Table 1. Characteristics of AIH patients with or without vitamin D deficiency.**

| | All (n = 66) | Total 25(OH)D <15 ng/mL (n = 36) | Total 25(OH)D ≥15 ng/mL (n = 30) | *P* value |
|---|---|---|---|---|
| Age, years, median (IQR) | 57 (50–66) | 57 (52–65) | 57 (46–66) | 0.6986 |
| Sex, male/female (female %) | 7/59 (89%) | 4/32 (89%) | 3/27 (90%) | 0.8839 |
| Laboratory data | | | | |
| PLT, ×10⁴/μL, median (IQR) | 16 (13–21) | 15 (14–20) | 16 (13–22) | 0.9920 |
| ALB, g/dL, median (IQR) | 3.5 (3.0–3.8) | 3.4 (2.8–3.7) | 3.8 (3.3–4.1) | 0.0050* |
| TB, mg/dL, median (IQR) | 2.0 (0.9–8.2) | 5.2 (1.5–15.9) | 1.1 (0.8–2.3) | 0.0007* |
| PT, % (IQR) | 74 (56–89) | 67 (54–83) | 85 (72–95) | 0.0048* |
| AST, U/L, median (IQR) | 240 (79–521) | 464 (120–670) | 119 (71–364) | 0.0341* |
| ALT, U/L, median (IQR) | 247 (98–815) | 352 (127–872) | 208 (96–665) | 0.1806 |
| ALP, U/L, median (IQR) | 399 (299–544) | 399 (344–534) | 387 (270–655) | 0.6520 |
| IgG, mg/dL median (IQR) | 2530 (1911–3171) | 2695 (1980–3082) | 2341 (1799–3178) | 0.3512 |
| ANA | | | | 0.6507 |
| < x 40 n (%) | 5 (7) | 2 (6) | 3 (10) | |
| x 40 n (%) | 1 (2) | 1 (3) | 0 (0) | |
| x 80 n (%) | 6 (9) | 3 (8) | 3 (10) | |
| > x 80 n (%) | 54 (82) | 30 (83) | 24 (80) | |
| Anti-SMA | | | | 0.3102 |
| Negative | 40 (61) | 24 (67) | 16 (53.3) | |
| x 40 n (%) | 14 (21) | 7 (19) | 7 (23.3) | |
| x 80 n (%) | 6 (9) | 2 (6) | 4 (13.3) | |
| > x 80 n (%) | 6 (9) | 3 (8) | 3 (10) | |
| Anti-LKM-1 | | | | 0.0448* |
| Index <17, (-) n (%) | 61 (92) | 31 (86) | 30 (100) | |
| Index 17–49, (±) n (%) | 4 (6) | 4 (11) | 0 (0) | |
| Index ≥50, (+) n (%) | 1 (2) | 1 (3) | 0 (0) | |
| 25(OH) D, ng/mL (IQR) | 14.2 (11.4–17.9) | 11.7 (9.6–13.0) | 18.3 (16.4–20.6) | <0.0001* |
| Scoring | | | | |
| Revised score, median (IQR) | 17 (15–19) | 18 (15–19) | 17(14–19) | 0.6066 |
| Simplified score median (IQR) | 7 (7–7) | 7 (7–7) | 7 (7–7) | 0.4122 |
| Definite AIH n (%) | 52 (79) | 29 (81) | 23 (77) | 0.7004 |
| Acute onset n (%) | 34 (52) | 23 (64) | 11 (37) | 0.0276* |
| Acute hepatitis phase n | 10 | 5 | 5 | |
| Acute exacerbation phase n | 24 | 18 | 6 | |
| Chronic onset n (%) | 32 (48) | 13 (36) | 19 (63) | |
| Severe form at diagnosis¶ n (%) | 20 (30) | 17 (47) | 3 (10) | 0.0011* |
| Cirrhosis at diagnosis n (%) | 10 (15) | 6 (17) | 4 (13) | 0.7069 |
| Staging of Fibrosis 0/1/2/3/4 (n = 61) | 6/9/19/19/8 | 3/3/9/12/6 | 3/6/10/7/2 | 0.0973 |
| Grading of activity 0/1/2/3/4 (n = 61) | 0/5/28/25/3 | 0/2/11/18/2 | 0/3/17/7/1 | 0.0322* |
| Complication of other autoimmune disease n (%) | 12 (18) | 6 (17) | 6 (20) | 0.7266 |
| Clinical outcomes | | | | |
| Relapse n (%) | 11 (18) | 8 (22) | 3 (10) | 0.1846 |
| Development of cirrhosis n (%) | 3 (5) | 3 (8) | 0 (0) | 0.1056 |
| Death or liver transplantation n (%) | 7 (11) | 5 (14) | 2 (7) | 0.3427 |
| Liver-related death or liver transplantation n (%) | 4 (6) | 3 (8) | 1 (3) | 0.3966 |
| Therapy | | | | |
| PSL therapy n (%) | 59 (89) | 33 (92) | 26 (87) | 0.5113 |
| Initial dose of 20–30 mg/day, n | 47 (79) | 25 (76) | 22 (85) | 0.2945 |

*(Continued)*

**Table 1.** (Continued)

|  | All (n = 66) | Total 25(OH)D <15 ng/mL (n = 36) | Total 25(OH)D ≥15 ng/mL (n = 30) | *P* value |
|---|---|---|---|---|
| Initial dose of 40–60 mg/day, n | 4 (7) | 2 (6) | 2 (8) |  |
| PSL pulse n (%) | 8 (14) | 6 (18) | 2 (8) |  |
| PSL/AZA combination therapy n (%) | 15 (23) | 12 (33) | 3 (10) | 0.0243* |
| Initial dose 50 mg/day, n | 13 (87) | 10 (83) | 3 (100) | 0.4475 |

*$P < 0.05$ was considered significant [25(OH)D, <15 ng/mL vs. ≥15 ng/mL].

¶Severe form: total bilirubin >5.0 mg/dL, and or prothrombin time <40%.

Abbreviations: ALB, albumin; TB, total bilirubin; AST, aspartate aminotransferase; ALT, alanine aminotransferase; PLT, platelet count; PT, Prothrombin time; PSL, predonisolone; AZA, azathioprine; ANA, antinuclear antibodies; anti-LKM-1, anti-liver/kidney microsomal 1; Anti-SMA, anti-smooth muscle antibodies; ALP, alkaline phosphatase; AIH, autoimmune hepatictis; IgG, immunoglobulin G; IQR, interquartile range; 25(OH)D, 25-hydroxyvitamin D.

serum total 25(OH)D levels of < 15 ng/mL than among patients with serum total 25(OH)D levels of ≥ 15 ng/mL.

### Patients with acute AIH have lower serum total 25-hydroxyvitamin D levels and higher percentage of free 25-hydroxyvitamin D than patients with chronic AIH

The median total 25(OH)D level among all AIH patients was 14.2 ng/mL (IQR, 11.4–17.9 ng/mL). The median level of total 25(OH)D in serum samples from patients with AIH was significantly lower than that in serum samples from age- and sex-matched healthy controls (HCs) (14.2 ng/ml vs. 18.2 ng/mL, $P = 0.0165$) (Fig 1A, S1 Table). When patients with acute-onset (n = 34) AIH at presentation were compared with those with chronic-onset (n = 32) AIH at presentation, the serum total 25(OH)D levels were found to be significantly lower in patients with acute AIH than in those with chronic AIH (13.2 ng/mL vs. 16.0 ng/mL, $P = 0.029$). In particular, serum total 25(OH)D levels were significantly lower in patients with severe forms of AIH than in patients with nonsevere forms (12.0 ng/mL vs. 16.0 ng/mL, $P = 0.0018$) (Fig 1B). We compared the serum total 25(OH)D levels between patients with acute AIH and patients with acute hepatitis due to other causes (11 with DILI and 3 with acute hepatitis B). The patients in these analyses were matched for age, sex and severity of liver injury with patients with acute AIH (S1 Table). No significant differences in the serum total 25(OH)D levels were found between patients with acute AIH and those with acute hepatitis of other etiologies (Fig 1A).

We also investigated serum free 25(OH)D levels in patients with AIH and in HCs (Fig 1C). The median level of free 25(OH)D in serum samples from patients with AIH (2.6 pg/mL) was not significantly different from that in serum samples from HCs (2.6 pg/mL, $P = 0.8471$). In addition, the serum free 25(OH)D levels did not differ significantly between patients with acute AIH and patients with chronic AIH (2.5 pg/mL vs. 2.9 pg/mL, $P = 0.4159$). Furthermore, we evaluated the percentage of free 25(OH)D (Free 25(OH)D/total 25(OH)D) in patients with AIH and in HCs (Fig 1D). The median percentage of free 25(OH)D in patients with AIH was significantly higher than that in HCs (0.018% vs. 0.016%, $P = 0.0496$). When patients with acute-onset AIH at presentation were compared with those with chronic-onset AIH at presentation, the percentage of free 25(OH)D was found to be significantly higher in patients with acute AIH than in patients with chronic AIH (0.023% vs. 0.016%, $P = 0.0106$). Fig 1E presents the relationships between serum free 25(OH)D and total 25(OH)D levels in patients with AIH and in HCs. Significant positive correlations between the free and total 25(OH)D levels were observed in each group (AIH: r = 0.4336, $P = 0.0003$; HC: r = 0.7697, $P = 0.0126$).

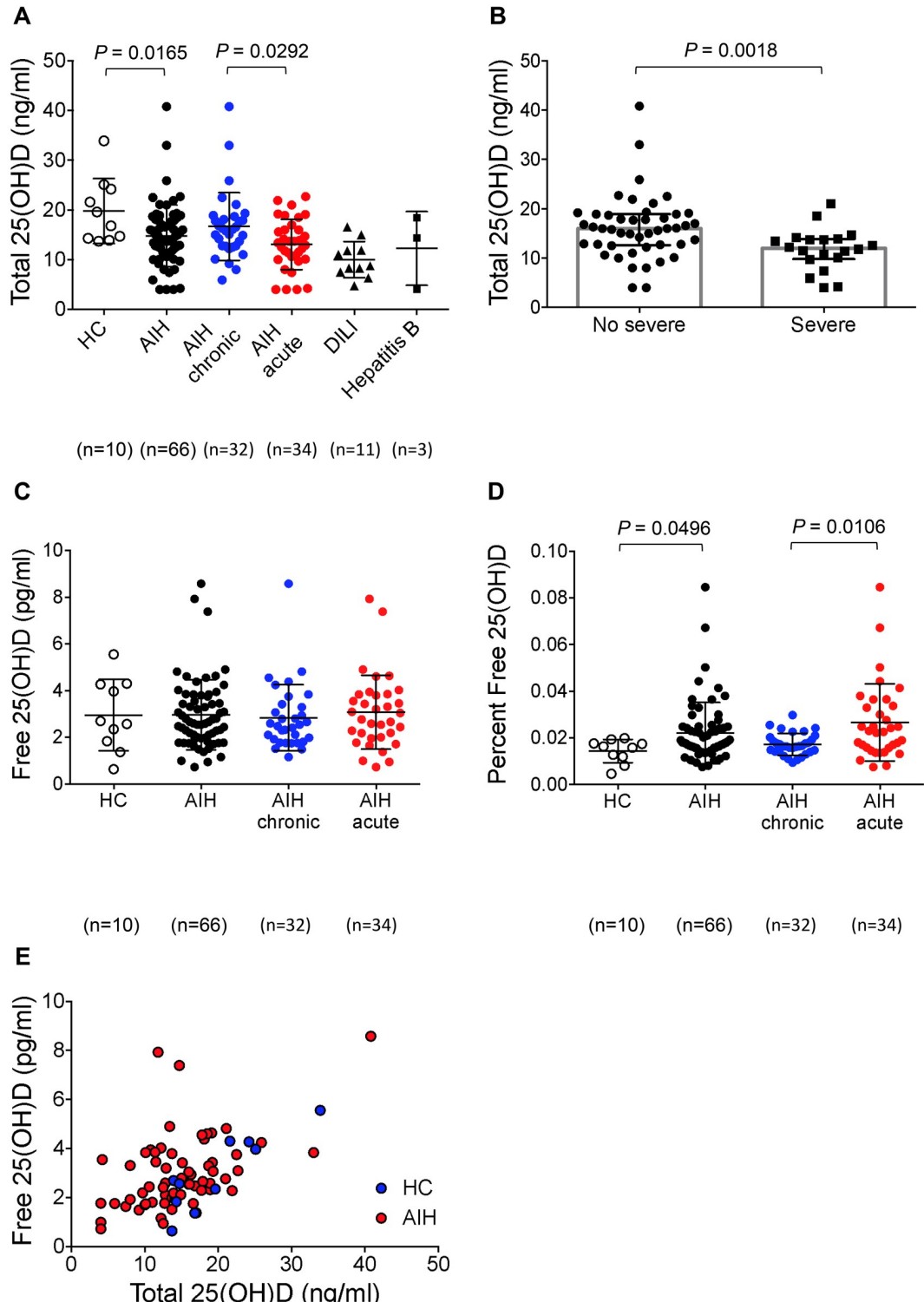

**Fig 1. Comparison of patients with acute- or chronic-onset AIH at presentation, DILI, or acute hepatitis B and HCs.** (A) Serum total 25(OH)D levels in AIH patients with acute-onset or chronic-onset disease at presentation, DILI, or acute hepatitis B and in HCs. (B) Comparison of serum total 25(OH)D levels in patients with severe and nonsevere forms of AIH. (C) Serum free 25(OH)D levels in AIH patients with acute-onset or chronic-onset disease at presentation and in healthy controls. (D) Percentage of free 25(OH)D in patients with AIH and in healthy controls. (E) Total 25(OH)D concentrations are plotted on the x-axis, and free 25(OH)D levels are plotted on the y-axis. The red circles represent data from patients with AIH, and the blue circles represent data from HCs. The horizontal line indicates the medians (IQRs). $P$ values were calculated with the Mann-Whitney U test; $P < 0.05$ was considered significant. Abbreviation: NS, nonsignificant.

**Table 2. Relationship between total 25-hydroxyvitamin D and clinical presentation in patients with AIH.**

| Variable | Total 25(OH)D | |
|---|---|---|
| | r | P |
| AST (U/L) | -0.2385 | 0.0619 |
| ALT (U/L) | -0.1816 | 0.1444 |
| ALP (U/L) | 0.0190 | 0.8796 |
| TB (mg/dL) | -0.3539 | 0.0038* |
| ALB (g/dL) | 0.3331 | 0.0093* |
| PT (%) | 0.3589 | 0.0036* |
| PLT (x10⁴/μL) | -0.0339 | 0.7903 |
| IgG (mg/dL) | -0.0381 | 0.7613 |
| Staging of fibrosis | -0.0799 | 0.5401 |
| Grading of activity | -0.4039 | 0.0012* |

*$P < 0.05$ was considered significant.

Abbreviations: ALB, albumin; ALP, alkaline phosphatase; ALT, alanine aminotransferase; AST, aspartate aminotransferase; IgG, immunoglobulin G; PLT, platelet count; PT, Prothrombin time; TB, total bilirubin; 25(OH)D, 25-hydroxyvitamin D.

## Relationship between total and free 25-hydroxyvitamin D levels and histological features

Although serum total 25(OH)D levels were not correlated with the stage of fibrosis (r = -0.0799; $P$ = 0.5401), they were moderately but significantly negatively correlated with the necroinflammatory activity grade (r = -0.4039; $P$ = 0.0012) (Table 2). Patients with AIH were also stratified by the progression of necroinflammatory activity (Fig 2). When serum total 25 (OH)D levels were compared between patients exhibiting severe necroinflammatory activity (G3-G4) and those exhibiting nonsevere necroinflammatory activity (G0-G2) as assessed by liver histology, the serum total 25(OH)D levels in patients with severe necroinflammatory

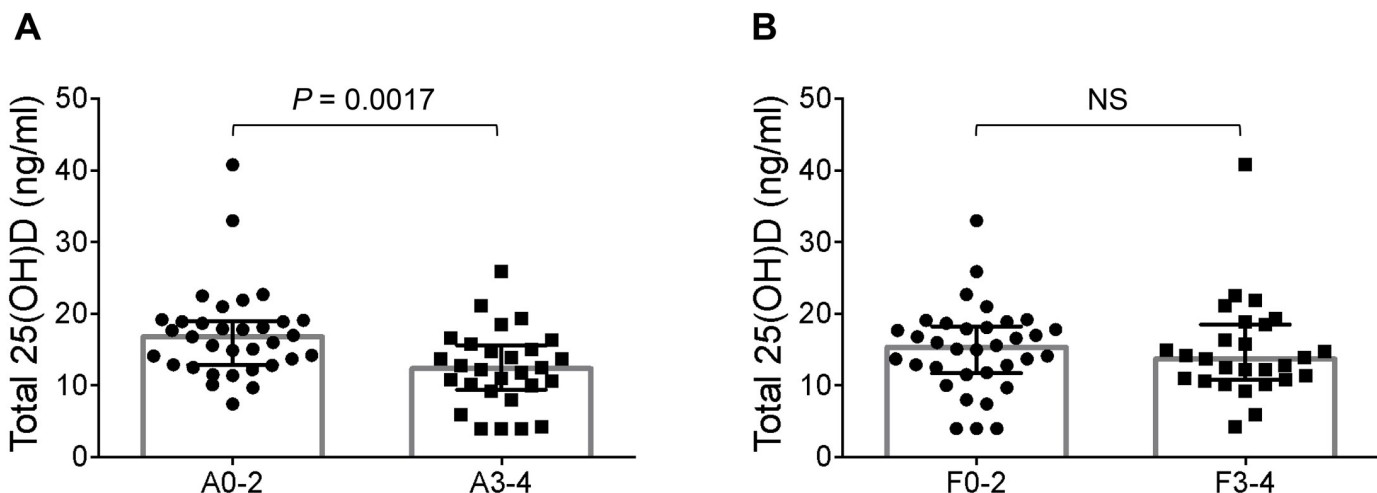

**Fig 2. Relationship between serum total 25(OH)D levels and histological features in patients with AIH.** Serum total 25(OH)D levels according to the degree of necroinflammatory activity and liver fibrosis. (A) Comparison of serum total 25(OH)D levels between patients with high and low necroinflammatory activity grades as assessed by liver histology. (B) Comparison of serum total 25(OH)D levels between patients with early and advanced stages of fibrosis as assessed by liver histology. The horizontal line indicates the medians (IQRs). $P$ values were calculated with the Mann-Whitney U test; $P < 0.05$ was considered significant.

activity were found to be significantly lower (12.4 ng/mL vs. 16.8 ng/mL; *P* = 0.0017) (Fig 2A and 2B). The results of univariate and multivariate logistic regression analyses for factors significantly associated with severe necroinflammatory activity are shown in Table 3. Multivariate analysis was performed with the factors identified as significant by univariate analysis; the total 25(OH)D level and IgG titer were found to be independent factors. The odds ratio (OR) for severe necroinflammatory activity with a low total 25(OH)D level was 0.86 (95% CI, 0.73–0.97; *P* = 0.0156). Although the serum total 25(OH)D levels in patients with DILI were not correlated with the serum ALT and TB levels or PT, they were significantly positively correlated with the serum ALB levels (r = 0.6324, *P* = 0.0403) (S2 Table). Furthermore, when the relationship between the percentage of free 25(OH)D and clinical presentation in patients with AIH was evaluated, the percentage of free 25(OH)D in patients with AIH was found to be significantly positively correlated with the serum TB level (r = 0.4239, *P* = 0.0004) and the activity grade (r = 0.4163, *P* = 0.0008) and negatively correlated with the serum PT (r = -0.3518, *P* = 0.0044) and ALB level (r = -0.3137, *P* = 0.0146) (S3 Table).

## Serum levels of inflammatory cytokines in AIH patients with serum total 25-hydroxyvitamin D levels of < 15 ng/mL or ≥ 15 ng/mL

Of the 66 patients with AIH, 30 had serum total 25(OH)D levels of ≥ 15 ng/mL, and 36 had serum total 25(OH)D levels of < 15 ng/mL. As shown in Table 4, the median serum levels of proinflammatory cytokines (IFN-γ and IL-33) were significantly higher in AIH patients with serum total 25(OH)D levels of < 15 ng/mL than in patients with serum total 25(OH)D levels of ≥ 15 ng/mL (IFN-γ: 0.21 vs. 0 pg/mL, *P* = 0.0181; IL-33: 26.54 vs. 13.13 pg/mL, *P* = 0.0036). However, the median serum levels of proinflammatory (IL-1β, IL-6, IL-17A, IL-17F, IL-21, IL-22, IL-23, IL-25, IL-31, TNF-α, and soluble CD40 ligand [sCD40L]) and anti-inflammatory (IL-4 and IL-10) cytokines did not differ significantly between the two groups. The percentage of free 25(OH)D in patients with AIH was significantly negatively correlated with the sCD40L level (r = -0.2628, *P* = 0.0330) but was not correlated with the levels of other proinflammatory and anti-inflammatory cytokines.

**Table 3. Univariate and multivariate analyses of factors associated with an activity grade of A3-A4.**

| A3-4 vs. A0-2 | Univariate | | Multivariate | |
| --- | --- | --- | --- | --- |
| | OR (95%CI) | *P* | OR (95%CI) | *P* |
| Age, (year) | 1.01 (0.97–1.05) | 0.9906 | 0.99 (0.94–1.05) | 0.8394 |
| Sex, male | 0.43 (0.06–2.19) | 0.3192 | 0.44 (0.05–3.17) | 0.4288 |
| AST (U/L) | 1.00 (0.99–1.00) | 0.1733 | | |
| ALT (U/L) | 1.00 (0.99–1.00) | 0.3352 | | |
| ALP (U/L) | 1.00 (0.99–1.00) | 0.6042 | | |
| TB (mg/dL) | 1.06 (0.99–1.14) | 0.0737 | | |
| ALB (g/dL) | 0.29 (0.10–0.73) | 0.0071* | 0.57 (0.15–1.94) | 0.3733 |
| PT (%) | 0.96 (0.93–0.99) | 0.0031* | 0.98 (0.94–1.02) | 0.4940 |
| PLT (x10^4/μL) | 0.96 (0.87–1.02) | 0.2292 | | |
| IgG (mg/dL) | 1.00 (1.00–1.00) | 0.0244* | 1.00 (1.00–1.00) | 0.0314* |
| Total 25(OH)D (ng/mL) | 0.84 (0.73–0.94) | 0.0011* | 0.86 (0.73–0.97) | 0.0156* |

*$P < 0.05$ was considered significant.

Abbreviations: ALB, albumin; TB, total bilirubin; AST, aspartate aminotransferase; ALT, alanine aminotransferase; PLT, platelet count; PT, Prothrombin time; ALP, alkaline phosphatase; IgG, immunoglobulin G; 25(OH)D, 25-hydroxyvitamin D; OR, odds ratio.

**Table 4. Comparison of serum cytokine levels among AIH patients with normal and low serum total 25-hydroxy-vitamin D.**

| Serum cytokine (pg/mL) | Total 25(OH) D <15 ng/mL (n = 36) | Total 25(OH) D ≥15 ng/mL (n = 30) | P value |
|---|---|---|---|
| IL-1β, median (IQR) | 0.12 (0.09–0.15) | 0.15 (0.09–0.22) | 0.0720 |
| IL-4, median (IQR) | 0 (0–0) | 0 (0–0) | 0.1647 |
| IL-6, median (IQR) | 4.97 (1.77–7.27) | 4.41 (2.50–8.77) | 0.4068 |
| IL-10, median (IQR) | 0 (0–0.85) | 0 (0–1.5) | 0.8254 |
| IL-17A, median (IQR) | 0.85 (0.51–1.08) | 0.63 (0.33–1.05) | 0.3947 |
| IL-17F, median (IQR) | 0 (0–0) | 0 (0–0) | 0.8351 |
| IL-21, median (IQR) | 0.65 (0–22.81) | 0.32 (0–5.78) | 0.3515 |
| IL-22, median (IQR) | 0 (0–0) | 0 (0–0.63) | 0.2000 |
| IL-23, median (IQR) | 4.56 (0–15.6) | 4.56 (4.56–26.41) | 0.2986 |
| IL-25, median (IQR) | ND | 0 (0–0) | - |
| IL-31, median (IQR) | 167.17 (0–293.16) | 172.83 (95.62–265.18) | 0.7488 |
| IL-33, median (IQR) | 26.54 (15.07–48.35) | 13.13 (0–22.74) | 0.0036* |
| IFN-γ, median (IQR) | 0.21 (0–2.37) | 0 (0–0.32) | 0.0181* |
| TNF-α, median (IQR) | 4.12 (3.08–7.55) | 5.12 (2.89–9.84) | 0.2710 |
| sCD40L, median (IQR) | 172.31 (95.81–246.87) | 192.30 (141.23–267.92) | 0.6065 |

*$P < 0.05$ was considered significant.

Abbreviations: 25(OH)D, 25-hydroxyvitamin D; IL, interleukin; IFN, interferon; TNF, tumor necrosis factor; IQR, interquartile range; sCD40L, soluble CD40 ligand; ND, not detected.

## Discussion

In the present study, we investigated the association of serum total and free 25(OH)D levels with severe necroinflammatory activity and inflammatory cytokine levels in patients with AIH. Serum 25(OH)D is critical for immunoregulatory, antioxidant, and antifibrotic actions in patients with AIH [7,27–30]. To the best of our knowledge, this report is the first to show that the serum total 25(OH)D level is an independent factor for severe necroinflammatory activity and is associated with the levels of inflammatory cytokines such as IFN-γ and IL-33 in patients with AIH. As 25(OH)D deficiency may play an important role in predicting AIH severity via inflammatory cytokine production, we believe that the results of our study have important implications for these patients.

Several reports have described the role of vitamin D in patients with AIH [7,10,27,31]. Low serum vitamin D levels have been observed in 81% of patients with AIH [10]. Furthermore, vitamin D deficiency has been associated with severe interface hepatitis, advanced hepatic fibrosis, nonresponse to conventional immunosuppressive therapy, and liver-related death or the need for liver transplantation in patients with AIH [10]. In the present study, we demonstrated that serum total 25(OH)D levels were significantly lower in patients with acute-onset AIH than in those with chronic-onset AIH. In particular, serum total 25(OH)D levels were significantly lower in patients with severe forms of AIH than in patients with nonsevere forms. Furthermore, total serum 25(OH)D levels were positively correlated with the serum ALB level and PT and negatively correlated with the serum TB level and necroinflammatory activity in patients with AIH. Multivariate analysis was performed with the factors identified as significant by univariate analysis, and the total 25(OH)D level and IgG titer were found to be independent factors for severe necroinflammatory activity in AIH. However, our analysis did not show a correlation between serum total 25(OH)D levels and the stage of fibrosis, because 50% of the AIH patients had acute-onset AIH and only 15% had cirrhosis.

On the other hand, in this study, the number of patients treated with azathioprine in combination with PSL was higher among patients with serum total 25(OH)D levels of < 15 ng/mL than among patients with serum total 25(OH)D levels of ≥ 15 ng/mL. Treatment with azathioprine in combination with PSL has been used for AIH patients who do not respond to conventional PSL monotherapy in Japan. As previously reported, vitamin D deficiency has been shown to be associated with nonresponse to conventional immunosuppressive therapy.

Additionally, the effects of vitamin D on the production and activity of several cytokines have recently been intensively investigated. Vitamin D regulates inflammatory cytokines, which contribute to immune signaling and host defense [32]. In vitro studies have shown that vitamin D can inhibit the production of the proinflammatory cytokines IL-17, IFN-γ, and IL-6 [32–35]. In addition, animal studies have shown that vitamin D suppresses the production of IL-6, IFN-γ, and TNF-α [32,36–38]. Although human studies investigating the relationships between vitamin D and inflammatory cytokines are scarce and have yielded inconsistent results, previous reports have demonstrated the implication of several cytokines in the pathogenesis and severity of AIH [32,39,40].

In this study, the median serum IFN-γ level was significantly higher in AIH patients with serum total 25(OH)D levels of < 15 ng/mL than in patients with serum total 25(OH)D levels of ≥ 15 ng/mL. Soluble liver antigen-specific IFN-γ responses were found to be significantly more frequent in AIH patients than in control individuals [41]. IFN-γ is positively correlated with transaminase levels and with diseases treated with immunosuppressive therapy or in remission [40,42,43]. Overexpression of IFN-γ-inducible protein 10 was identified in the livers of patients with type I AIH [42,44]. In addition, IFN-γ production was required for pathogenesis in a murine model of fulminant liver inflammation and a murine model of hepatitis [42,45,46]. Moreover, calcitriol decreases serum ALT levels, markedly attenuates histological liver damage, and causes a reduction in the IFN-γ level in concanavalin A-induced hepatitis [42,47]. Calcitriol inhibits CD40-induced IFN-γ and immunomodulatory activity in human monocytes [42,48] and is a potent suppressor of IFN-γ-mediated macrophage activation [42,49]. Further, the ability of calcitriol to inhibit the production of IFN-γ in patients with AIH should be validated [42].

On the other hand, IL-33 was recently described as a new member of the IL-1 family, whose members exhibit proinflammatory activity [26]. Following cellular stress or damage, IL-33 is released in either its full length or cleaved form [26]. However, unlike IL-1β, IL-33 is not cleaved by caspase-1, and its cleavage is not necessary for its secretion, biological activity, or release, further suggesting that IL-33 plays a role as an alarmin [26]. IL-33 is released by damaged or necrotic cells, leading to activation of the immune system through IL-33 signaling [26,50]. Our previous report revealed that serum IL-33 levels were higher in patients with AIH than in control individuals [26]. Additionally, serum IL-33 levels were significantly higher in patients with AIH with acute presentation than in those with chronic presentation [26]. Furthermore, elevated levels of serum IL-33 were positively correlated with liver injury, as indicated by the levels of ALT and TB and the grade of necroinflammatory activity [26]. Multivariate analysis with the factors identified as significant by univariate analysis showed that the serum IL-33 level was an independent factor for severe necroinflammatory activity [26]. In another study, the serum IL-33 levels in patients with acute-onset AIH were positively correlated with hypergammaglobulinemia, liver injury, and proinflammatory cytokine levels [26,51]. In the present study, the median serum IL-33 level was significantly higher in AIH patients with serum total 25(OH)D levels of < 15 ng/mL than in those with serum total 25 (OH)D levels of ≥ 15 ng/mL. Further research on the systemic and localized effects of IL-33/ 25(OH)D in patients with AIH will provide a basis for targeted therapy that could benefit these patients.

Conversely, in this study, the serum free 25(OH)D levels in patients with AIH and HCs were similar despite the lower total 25(OH)D levels in patients with AIH. In a healthy individual, approximately 0.03% of 25(OH)D is in a free form; 85% is bound to VDBP, and 15% is bound to ALB [52]. Liver diseases associated with impaired protein synthetic function, such as cirrhosis and acute liver failure, result in reductions in VDBP and ALB levels [52]. In this study, the correlation between the free 25(OH)D and total 25(OH)D levels was significantly steeper in patients with AIH than in HCs, indicating that the affinity of VDBP and ALB for 25(OH)D was altered. Furthermore, the percentage of free 25(OH)D in patients with AIH was found to be significantly positively correlated with the serum TB level and the activity grade but negatively correlated with the serum PT and ALB levels. Patients with acute AIH and protein synthesis dysfunction have a higher percentage of free 25(OH)D than patients with chronic AIH without protein synthesis dysfunction, but the free 25(OH)D levels are similar because of the lower total 25(OH)D levels in patients with AIH [53]. Furthermore, correlation analysis of the percentage of free 25(OH)D with inflammatory factors in patients with AIH showed that the percentage of free 25(OH)D in patients with AIH was negatively correlated with the level of sCD40L. The CD40-CD40L costimulatory pathway is involved in the evolution of many autoimmune diseases, including systemic lupus erythematosus (SLE), rheumatoid arthritis (RA) and Sjögren's syndrome (SS) [54]. Increased serum levels of sCD40L have been associated with disease activity in autoimmune diseases [54].

Our study has several limitations. First, the sample population was relatively small. Second, we used a retrospective design; thus, our results need confirmation in a prospective study. Third, we should validate the predictive ability of total 25(OH)D in other cohorts to confirm our results. In addition, no relevant levels of cytokines were detected in this study, although the cytokines were detected or the levels were unchanged in only a few samples. Since the serum was preserved, it exhibited reduced cytokine activity. In addition, acute-onset AIH has many causes, and there may be differences in the cytokine networks. Moreover, the sensitivity of the measurement kit may be limited.

Collectively, these results indicate that vitamin D deficiency is associated with necroinflammatory activity in AIH, particularly via an increase in inflammatory cytokine production. Additional clinical studies with 25(OH)D need to be conducted to determine the mechanism by which 25(OH)D suppresses the pathogenesis of AIH.

## Supporting information

**S1 Table. Characteristics of patients with AIH, DILI, acute hepatitis B, and healthy individuals.**
(DOCX)

**S2 Table. Relationship between total 25-hydroxyvitamin D and clinical presentation in patients with DILI.**
(DOCX)

**S3 Table. Relationship between the percentage of free 25-hydroxyvitamin D and clinical presentation in patients with AIH.**
(DOCX)

**S4 Table. Relationship between the percentage of free 25-hydroxyvitamin D and cytokine profile in patients with AIH.**
(DOCX)

## Acknowledgments

The authors thank Drs. Osamu Suzuki and Yuko Hashimoto (Department of Diagnostic Pathology, Fukushima Medical University) for assistance in histopathological diagnosis and Chikako Saito and Rie Hikichi for technical assistance.

## Author Contributions

**Conceptualization:** Hiromasa Ohira.

**Data curation:** Kazumichi Abe, Masashi Fujita, Manabu Hayashi, Atsushi Takahashi.

**Formal analysis:** Kazumichi Abe.

**Funding acquisition:** Hiromasa Ohira.

**Investigation:** Kazumichi Abe.

**Writing – original draft:** Kazumichi Abe.

**Writing – review & editing:** Hiromasa Ohira.

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
