## [Decision Letter · Decision Letter 0]

1 Jul 2020

PONE-D-20-17668

Association of Serum 25-Hydroxyvitamin D Levels with Severe Necroinflammatory Activity and Inflammatory Cytokine Production in Type I Autoimmune Hepatitis

PLOS ONE

Dear Dr. Abe,

Thank you for submitting your manuscript to PLOS ONE. After careful consideration, we feel that it has merit but does not fully meet PLOS ONE’s publication criteria as it currently stands. Therefore, we invite you to submit a revised version of the manuscript that addresses the points raised during the review process.

As you can see, both reviewers appreciated your work, both also highlighted several important issues, that need to be addressed.

We look forward to receiving your revised manuscript.

Kind regards,

Pavel Strnad

Academic Editor

PLOS ONE

Journal Requirements:

2. We noticed minor instances of text overlap with the following previous publication(s), which need to be addressed:

(1) https://aasldpubs.onlinelibrary.wiley.com/doi/full/10.1002/hep4.1326 (lines 273-296).

(2) https://www.mdpi.com/2072-6643/6/1/221/htm (lines 250-256)

(3) https://www.ncbi.nlm.nih.gov/pmc/articles/PMC3808258/ (lines 261-271)

The text that needs to be addressed involves the Discussion section.

In your revision please ensure you cite all your sources (including your own works), and quote or rephrase any duplicated text outside the methods section. Further consideration is dependent on these concerns being addressed.

"This study was supported in part by AbbVie Inc."

"Kazumichi Abe and Hiromasa Ohira has received research funds from AbbVie Inc. All other authors declare no conflict of interest."

We note that you received funding from a commercial source: AbbVie Inc.

Reviewers' comments:

Reviewer's Responses to Questions

**Comments to the Author**

1. Is the manuscript technically sound, and do the data support the conclusions?

Reviewer #1: Partly

Reviewer #2: Yes

2. Has the statistical analysis been performed appropriately and rigorously? 

Reviewer #1: Yes

Reviewer #2: Yes

3. Have the authors made all data underlying the findings in their manuscript fully available?

Reviewer #1: Yes

Reviewer #2: Yes

4. Is the manuscript presented in an intelligible fashion and written in standard English?

Reviewer #1: Yes

Reviewer #2: Yes

5. Review Comments to the Author

Reviewer #1: In their study entitled “Association of Serum 25-Hydroxyvitamin D Levels with Severe Necroinflammatory Activity and Inflammatory Cytokine Production” Abe et al. retrospectively analysed a single-centre cohort of AIH patients with regard to their 25-hydroxyvitamin D (25(OH)D) levels. The authors found higher levels of 25(OH)D in AIH patients with acute severe presentation, in line with a positive correlation between 25(OH)D levels and serum albumin and prothrombin time and a negative correlation with serum total bilirubin levels and histological grade of necroinflammatory activity. In addition, the author analysed several pro- and anti-inflammatory cytokines out of the sera of AIH patients and found higher levels of interferon-gamma and IL-33 levels in patients with low levels of 25(OH)D.

I have several major and minor concerns which need to be addressed before this study can be accepted for publication.

Major concerns:

1. My major concern lies in doubts that 25(OH)D plays a role in the pathogenesis of AIH as the authors claim it does. The low levels of 25(OH)D in patients with acute-severe presentation of AIH may be just a result of acute liver damage and of impaired liver function. The low levels of 25(OH)D could depend only on the severity of liver damage, but not on the kind liver disease. 25(OH)D is mainly bound to albumin and vitamin D binding protein in blood, and both transporters are produced in the liver. Due to acute liver injury (of any cause), production of albumin and vitamin D binding protein could be impaired and thereby lower levels of 25(OH)D were measured in peripheral blood of patients with acute presentation of AIH.

a. The authors must add a control groups to their study in order to support that lower levels of 25(OH)D are part of the pathogenesis of AIH. Can the authors perform measurements of 25(OH)D in patients with other causes of acute hepatitis (drug-induced liver injury, acute viral hepatitis)? These analyses should be matched to age, sex and severity of liver injury (e.g. matched to total bilirubin levels)

b. Can the authors also add measurements in age- and sex-matched healthy individuals?

c. What was measured by the 25(OH)D assay used in this study? Protein-bound or free 25(OH)D? Can the authors add measurements of free 25(OH)D, if this has not been performed yet?

2. What were the diagnostic criteria for AIH patients with acute presentation? The authors only mention that for the diagnosis of chronic AIH, the IAIHG and the simplified score were applied and that alcohol consumption and chronic viral hepatitis B and C infection were excluded. Did the authors also apply these diagnostic scores for AIH patients with acute presentation? How were important differential diagnoses of acute hepatitis excluded, such as drug-induced liver injury or acute viral hepatitis? Was acute viral hepatitis E infection excluded by PCR?

3. As I understand it right, 25(OH)D and laboratory values of patients with acute presentation of AIH were analysed at the time point of initial presentation. But at which time point were 25(OH)D and laboratory values analysed in patients with chronic manifestation of AIH? At the time of first diagnosis? Besides, the authors performed analyses at the time of remission. However, the time interval until biochemical remission is achieved, varies from patient to patient. Can the authors give the information what the median time was until remission was achieved in their cohort of AIH patients?

4. Many patients were treated with steroids (prednisolone monotherapy in 89% and combination therapy with azathioprine in 23% of patients). It seems that no patient with chronic AIH was on azathioprine monotherapy. Is this correct? At least those patients with chronic AIH analysed at the time point of remission, being treated for a longer period with steroids, should have been supplemented with vitamin D. How many patients were supplemented with vitamin D when measurements of vitamin D were performed?

5. Throughout the manuscript, the exact median results should be presented and not only p-values (e.g. page 12: “In particular, serum 25(OH)D levels were significantly lower in patients with severe symptoms of AIH than in patients with nonsevere symptoms (P = 0.0018)” – please present the exact levels of 25(OH)D!)

6. Page 12: „…but significantly positively correlated with the necroinflammatory activity grade (r = -0.4039)“ – “r“ is negative, so it is a negative correlation!

Minor concerns:

7. The sex distribution of this study differs from previous international reports, with far more female than male AIH patients in the study by Abe et al. (89% were female) than in previous international studies (about 70% of AIH patients were female). What is the authors’ explanation for this? What is the sex distribution of AIH patients in Japan due to population-based studies?

8. The statement “We investigated the association of serum 25(OH)D levels with pathogenesis” must be avoided: How can laboratory values be associated with pathogenesis itself? In this study, 25(OH)D levels were associated with cytokine levels. Furthermore, pathogenesis of AIH is still unclear and only certain aspects such as proinflammatory mechanisms have been investigated yet.

9. Page 3/4: The incidence of AIH with acute presentation has recently increased, and this form has been established a type I AIH” – this is not correct, paediatric patients with type 2 AIH (anti-LKM+) often manifest with acute and severe hepatitis.

10. In the methods section the authors state that anti-LKM were detected by ELISA. Which test was performed for the detection of ANA? Immunofluorescence or ELISA? What about the other antibodies that should be tested for in a patient suspected to have AIH (Anti-SMA? Anti-SLA/LP? Anti-LC1?)

11. Page 10: “Twenty patients (30%) had severe disease symptoms (TB >5.0 mg/dL and/or PT <40%)” – A laboratory value is not a symptom. Do you mean that 30% presented with icterus?

Reviewer #2: The manuscript reports an association between reduced vitamin D serum levels and necroinflammatory activity / disease severity as well as with some proinflammatory cytokines (INF-gamma and IL-33) in patients with autoimmune Hepatitis. The manuscript is well written and the analyses are thoroughly performed. The novelty of the study is - however - limited, yet the association of Vitamin D deficiency with IL-33 serum levels of some interest. Yet, careful proofreading should be performed in order to eliminate some typos / grammatical issues, and authors should explain or at least discuss why a relevant number of cytokines was not detectable at all.

6. PLOS authors have the option to publish the peer review history of their article (what does this mean?). If published, this will include your full peer review and any attached files.

Reviewer #1: No

Reviewer #2: No

---

## [Author Response · Author response to Decision Letter 0]

11 Aug 2020

Responses to the reviewers’ comments.

Thank you for inviting us to submit a revised draft of our manuscript entitled “Association of Serum 25-Hydroxyvitamin D Levels with Severe Necroinflammatory Activity and Inflammatory Cytokine Production in Type I Autoimmune Hepatitis” to PLOS ONE. We also appreciate the time and effort that you and each of the reviewers have dedicated to providing insightful feedback on ways to strengthen our paper. Thus, it is with great pleasure that we resubmit our article for further consideration. We have incorporated changes that reflect the detailed suggestions that you have graciously provided. We also hope that our edits and the responses that we provide below satisfactorily address all issues and concerns that you and the reviewers have noted.

Reviewer #1 

Major concerns:

1. My major concern lies in doubts that 25(OH)D plays a role in the pathogenesis of AIH as the authors claim it does. The low levels of 25(OH)D in patients with acute-severe presentation of AIH may be just a result of acute liver damage and of impaired liver function. The low levels of 25(OH)D could depend only on the severity of liver damage, but not on the kind liver disease. 25(OH)D is mainly bound to albumin and vitamin D binding protein in blood, and both transporters are produced in the liver. Due to acute liver injury (of any cause), production of albumin and vitamin D binding protein could be impaired and thereby lower levels of 25(OH)D were measured in peripheral blood of patients with acute presentation of AIH.

a. The authors must add a control groups to their study in order to support that lower levels of 25(OH)D are part of the pathogenesis of AIH. Can the authors perform measurements of 25(OH)D in patients with other causes of acute hepatitis (drug-induced liver injury, acute viral hepatitis)? These analyses should be matched to age, sex and severity of liver injury (e.g. matched to total bilirubin levels)

RESPONSE: We agree with you. We have included a new figure (Fig 1A) with the data for serum 25(OH)D levels in 14 patients with acute hepatitis (11 with drug-induced liver injury and 3 with acute hepatitis B). The patients in these analyses were matched for age, sex and severity of liver injury with patients with acute AIH (Table S1). We also investigated the relationship between total 25(OH)D and clinical presentation in patients with DILI (Table S2). (P14 L216, P17 L257, Fig 1A, Table S1, and Table S2).

b. Can the authors also add measurements in age- and sex-matched healthy individuals?

RESPONSE: We agree with you. We have included a new figure (Fig 1A) with the data for serum 25(OH)D levels in age- and sex-matched healthy individuals (P13 L208, Fig 1A, and Table S1).

c. What was measured by the 25(OH)D assay used in this study? Protein-bound or free 25(OH)D? Can the authors add measurements of free 25(OH)D, if this has not been performed yet?

RESPONSE: Thank you for your suggestion. In this study, we measured the total 25(OH)D levels in patients with AIH. We have included new figures (Fig 1C-E) with the data for serum free 25(OH)D levels in 66 patients with AIH and 10 healthy controls. We also investigated the percentage of free 25(OH)D and the relationship between the percentage of free 25(OH)D and clinical presentation in patients with AIH (P14 L223, P17 L260, P23 L363, Fig 1C-E, Table S3, and Table S4).

2. What were the diagnostic criteria for AIH patients with acute presentation? The authors only mention that for the diagnosis of chronic AIH, the IAIHG and the simplified score were applied and that alcohol consumption and chronic viral hepatitis B and C infection were excluded. Did the authors also apply these diagnostic scores for AIH patients with acute presentation? How were important differential diagnoses of acute hepatitis excluded, such as drug-induced liver injury or acute viral hepatitis? Was acute viral hepatitis E infection excluded by PCR?

RESPONSE: We agree with your assessment. Patients with an ALT level of at least 10 times the upper limit of normal and/or a TB level of at least 5 mg/dl were diagnosed with an acute-onset type, per ref. 26. Acute AIH was evaluated using the IAIHG score and a simplified score. We excluded patients with EBV, CMV, HAV, and HEV by testing for IgM-EBV, IgM-CMV, IgM-HAV, and IgA-HEV antibodies (Ab). We excluded patients with DILI by assessing the Digestive Disease Week Japan (DDW-J) 2004 score and liver biopsy findings (P6 L92, ref. 19, and ref. 26).

3. As I understand it right, 25(OH)D and laboratory values of patients with acute presentation of AIH were analyzed at the time point of initial presentation. But at which time point were 25(OH)D and laboratory values analyzed in patients with chronic manifestation of AIH? At the time of first diagnosis? Besides, the authors performed analyses at the time of remission. However, the time interval until biochemical remission is achieved, varies from patient to patient. Can the authors give the information what the median time was until remission was achieved in their cohort of AIH patients?

RESPONSE: Thank you for your suggestion. Serum 25(OH)D levels and laboratory values were analyzed in patients 

with chronic AIH at the time of first diagnosis. The median time to remission was 63 days (IQR, 43-108 days). Two patients did not achieve remission (P7 L108 and P7 L112).

4. Many patients were treated with steroids (prednisolone monotherapy in 89% and combination therapy with azathioprine in 23% of patients). It seems that no patient with chronic AIH was on azathioprine monotherapy. Is this correct? At least those patients with chronic AIH analyzed at the time point of remission, being treated for a longer period with steroids, should have been supplemented with vitamin D. How many patients were supplemented with vitamin D when measurements of vitamin D were performed?

RESPONSE: Thank you for providing these insights. Although corticosteroids are used as the first-line therapy for AIH, approximately 10% of AIH patients in Japan are resistant to steroid therapy. Steroid-resistant patients, although the definition is unclear, include patients whose transaminase levels do not decrease below two times the upper limit of normal after 6 months of regular steroid therapy, those for whom steroid therapy cannot be continued due to toxicity, those who experience exacerbation of hepatitis and those with liver failure not responsive to high-dose steroid therapy. Azathioprine, which was not covered by national health insurance in Japan until July 2018, is considered the second-line therapy for steroid-resistant patients. None of the AIH patients were taking vitamin D supplements when the serum 25(OH)D levels were measured. Six patients took vitamin D supplements after treatment (P8 L123 and P8 L126).

5. Throughout the manuscript, the exact median results should be presented and not only p-values (e.g. page 12: “In particular, serum 25(OH)D levels were significantly lower in patients with severe symptoms of AIH than in patients with nonsevere symptoms (P = 0.0018)” – please present the exact levels of 25(OH)D!)

RESPONSE: Thank you for your suggestion. We have revised the text (P14 L210, P14 L214, P14 L216, P15 L226, P15 L228, P15 L231, P15 L234, P15 L237, P16 L244, P16 L245, P16 L251, P17 L259 and P17 L263).

6. Page 12: „…but significantly positively correlated with the necroinflammatory activity grade (r = -0.4039)“ – “r“ is negative, so it is a negative correlation!

RESPONSE: We agree with you. We have revised the text (P16 L244).

Minor concerns:

7. The sex distribution of this study differs from previous international reports, with far more female than male AIH patients in the study by Abe et al. (89% were female) than in previous international studies (about 70% of AIH patients were female). What is the authors’ explanation for this? What is the sex distribution of AIH patients in Japan due to population-based studies?

RESPONSE: Thank you for providing these insights. The male:female sex distribution among AIH patients in Japan is reported to be 1:7 (manuscript ref. 3). However, reports from a single facility or from each region indicate a ratio of 1:8-9 (1-2). In addition, the population of patients with acute-onset AIH in Japan is reported to be composed of 87% women (3). In this study, more than half of the originally identified acute-onset AIH patients were included, suggesting that the proportion of women may be high.

1. Miyake Y, Iwasaki Y, Sakaguchi K, Shiratori Y. Clinical features of Japanese male patients with type 1 autoimmune hepatitis. Aliment Pharmacol Ther. 2006 Aug 1;24(3):519-23. doi: 10.1111/j.1365-2036.2006.03013.x.PMID: 16886918 

2. Takahashi A, Ohira H, Abe K, Miyake Y, Abe M, Yamamoto K, Suzuki Y, Onji M, Tsubouchi H; Intractable Liver and Biliary Diseases Study Group of Japan. Rapid corticosteroid tapering: Important risk factor for type 1 autoimmune hepatitis relapse in Japan. Hepatol Res. 2015 Jun;45(6):638-44. doi: 10.1111/hepr.12397. Epub 2014 Aug 27.PMID: 25070037

3. Joshita S, Yoshizawa K, Umemura T, Ohira H, Takahashi A, Harada K, Hiep NC, Tsuneyama K, Kage M, Nakano M, Kang JH, Koike K, Zeniya M, Yasunaka T, Takaki A, Torimura T, Abe M, Yokosuka O, Tanaka A, Takikawa H; Japan Autoimmune Hepatitis Study Group (JAIHSG). Clinical features of autoimmune hepatitis with acute presentation: a Japanese nationwide survey. J Gastroenterol. 2018 Sep;53(9):1079-1088. doi: 10.1007/s00535-018-1444-4. Epub 2018 Feb 23.PMID: 29476251 

8. The statement “We investigated the association of serum 25(OH)D levels with pathogenesis” must be avoided: How can laboratory values be associated with pathogenesis itself? In this study, 25(OH)D levels were associated with cytokine levels. Furthermore, pathogenesis of AIH is still unclear and only certain aspects such as proinflammatory mechanisms have been investigated yet.

RESPONSE: We agree with you. We have revised the manuscript (P2 L21 and P5 L79).

9. Page 3/4: The incidence of AIH with acute presentation has recently increased, and this form has been established a type I AIH” – this is not correct, paediatric patients with type 2 AIH (anti-LKM+) often manifest with acute and severe hepatitis.

RESPONSE: We agree with you. We have revised the manuscript (P4 L52).

10. In the methods section the authors state that anti-LKM were detected by ELISA. Which test was performed for the detection of ANA? Immunofluorescence or ELISA? What about the other antibodies that should be tested for in a patient suspected to have AIH (Anti-SMA? Anti-SLA/LP? Anti-LC1?)

RESPONSE: Thank you for your suggestion. We have added the anti-smooth muscle antibody (anti-SMA) data. Autoantibodies (antinuclear antibodies [ANAs] and anti-smooth muscle antibodies [anti-SMAs]) were detected by indirect immunofluorescence in HEp-2 cells or frozen sections of rat kidney tissue (P6 L86 and Table 1).

11. Page 10: “Twenty patients (30%) had severe disease symptoms (TB >5.0 mg/dL and/or PT <40%)” – A laboratory value is not a symptom. Do you mean that 30% presented with icterus?

RESPONSE: We agree with you. We have revised the manuscript (P2 L28, P8 L121, P12 L182, P19 L300, and the legend for Fig 1B).

Reviewer #2: The manuscript reports an association between reduced vitamin D serum levels and necroinflammatory activity / disease severity as well as with some proinflammatory cytokines (INF-gamma and IL-33) in patients with autoimmune Hepatitis. The manuscript is well written and the analyses are thoroughly performed. The novelty of the study is - however - limited, yet the association of Vitamin D deficiency with IL-33 serum levels of some interest. Yet, careful proofreading should be performed in order to eliminate some typos / grammatical issues, and authors should explain or at least discuss why a relevant number of cytokines was not detectable at all.

RESPONSE: We agree with you, and we have addressed this suggestion throughout our manuscript. Since the serum was preserved, it exhibited reduced cytokine activity. Acute-onset AIH has many causes, and there may be differences in the cytokine networks Moreover, the sensitivity of the measurement kit may be limited. We have added this text to the Discussion section (P25 L387).

---

## [Decision Letter · Decision Letter 1]

7 Sep 2020

Association of Serum 25-Hydroxyvitamin D Levels with Severe Necroinflammatory Activity and Inflammatory Cytokine Production in Type I Autoimmune Hepatitis

PONE-D-20-17668R1

Dear Dr. Abe,

We’re pleased to inform you that your manuscript has been judged scientifically suitable for publication and will be formally accepted for publication once it meets all outstanding technical requirements.

Kind regards,

Pavel Strnad

Academic Editor

PLOS ONE

Additional Editor Comments (optional):

Reviewers' comments:

Reviewer's Responses to Questions

**Comments to the Author**

1. If the authors have adequately addressed your comments raised in a previous round of review and you feel that this manuscript is now acceptable for publication, you may indicate that here to bypass the “Comments to the Author” section, enter your conflict of interest statement in the “Confidential to Editor” section, and submit your "Accept" recommendation.

Reviewer #2: All comments have been addressed

2. Is the manuscript technically sound, and do the data support the conclusions?

Reviewer #2: Yes

3. Has the statistical analysis been performed appropriately and rigorously? 

Reviewer #2: Yes

4. Have the authors made all data underlying the findings in their manuscript fully available?

Reviewer #2: Yes

5. Is the manuscript presented in an intelligible fashion and written in standard English?

Reviewer #2: Yes

6. Review Comments to the Author

Reviewer #2: The authors have sufficiently responded to my concerns. As mentioned previously, the analysis have been well performed and the manuscript is interesting to read, although the data are rather of confirmatory nature.

7. PLOS authors have the option to publish the peer review history of their article (what does this mean?). If published, this will include your full peer review and any attached files.

Reviewer #2: No

---

## [Editor Report · Acceptance letter]

27 Oct 2020

PONE-D-20-17668R1 

Association of Serum 25-Hydroxyvitamin D Levels with Severe Necroinflammatory Activity and Inflammatory Cytokine Production in Type I Autoimmune Hepatitis 

Dear Dr. Abe:

I'm pleased to inform you that your manuscript has been deemed suitable for publication in PLOS ONE. Congratulations! Your manuscript is now with our production department. 

Kind regards, 

on behalf of

Dr. Pavel Strnad 

Academic Editor

PLOS ONE